# Co-design to consensus: Identifying the core elements of a novel intervention for pre-school children with co-occurring phonological speech sound disorder (SSD) and developmental language disorder (DLD) using a modified e-Delphi approach

Lucy Rodgers[1,2*], Nicola Botting[1], Helen Stringer[3], Natalie Abdo[2], Meriem Amer-El-Khedoud[1,4], Emma Baker[5], Sophie Franks[6], Dave Harford[7], Patrycja Salimi-Tabar[8], Laura Temple[9], Ros Herman[1]

**1** Department of Language and Communication Sciences, City St George's, London, United Kingdom, **2** Sussex Community NHS Foundation Trust, Brighton, United Kingdom, **3** School of Education, Communication and Language Sciences, Newcastle University, Newcastle, United Kingdom, **4** Barts Health NHS Trust, London, United Kingdom, **5** Sheffield Children's NHS Foundation Trust, Sheffield, United Kingdom, **6** West Sussex County Council, Worthing, United Kingdom, **7** Birmingham City university, Faculty of Health, Education and Life Sciences, Birmingham, United Kingdom, **8** Ethnic Minority Achievement Service, Brighton and Hove City Council, Brighton, United Kingdom, **9** Brighton and Hove Inclusion Support Service, Brighton and Hove City Council, Brighton, United Kingdom

* lucy.rodgers@city.ac.uk

## Abstract

### Introduction

Although frequently seen in clinical services, there are few interventions which have been developed specifically to meet the needs of pre-school children with co-occurring features of a phonological speech sound disorder (P-SSD) and developmental language disorder (DLD). This study aims to achieve consensus on the core elements of a novel intervention for pre-school children with co-occurring features of P-SSD and DLD ("SWanS"- Supporting Words and Sounds), where expressive vocabulary and speech comprehensibility are joint outcomes of interest.

### Methods

Forty-seven potential core intervention elements, based on a priori findings and the wider literature, were generated by a diverse steering group of professionals and people with lived experience within a systematic co-design process. This was followed by a modified, two round, e-Delphi with expert Speech and Language Therapists (SLTs) to achieve consensus on the elements. Consensus was defined as over 75% of participants (minimum 30 SLTs) rating the elements as either appropriate or very appropriate on a Likert of 1–5, with an inter-quartile range of one or below. If

**Data availability statement:** 1. All relevant data for the steering group co-design process are contained within the Supporting Information files. 2. All raw e-Delphi data are publically available at: Rodgers, Lucy (2024). Anonymised raw data (e-delphi). City, University of London. Dataset. https://doi.org/10.25383/city.27623454.v1

**Funding:** This work was supported by the Wellcome Trust via a Health Advances in Underrepresented Populations and Diseases (HARP) Fellowship, awarded to Lucy Rodgers. Grant no. 223500/Z/21/Z. There was no additional external funding received for this study.

**Competing interests:** The authors have declared that no competing interests exist.

consensus was not achieved in round 1, free text comments were used to generate amended statements for the second round.

## Results

Consensus was achieved on 42/47 statements in round 1. During the revision process, one statement was discarded; six statements which did not achieve consensus were re-worded; two statements which required further clarity had examples added; four statements were merged into two statements. Consensus was reached on 8/8 statements presented in round 2, resulting in 44 final statements achieving consensus in total.

## Conclusions

Core elements of a novel intervention have been identified through co-design with a diverse group of stakeholders followed by consensus with expert SLTs. Additional flexibility was required within some core elements in order to achieve consensus. Implications for future implementation are discussed.

## Introduction

Early speech and language difficulties can significantly impact a child's wellbeing and quality of life [1,2]. However, communication difficulties in early childhood can be ameliorated, or even resolved, by access to speech and language therapy intervention. Speech and language therapy interventions are typically developed for children with singular profiles [3], despite many children having characteristics of more than one type of difficulty [4]. This study concerns identifying the core elements of a novel, complex, theory-based combined intervention for pre-school children with a co-occurring profile of developmental language disorder (DLD) and phonological speech sound disorder (P-SSD), named "SWanS" (Supporting Words and Sounds). Following the co-design of potential intervention elements with a diverse project steering group, consensus on these elements was sought using e-Delphi methodology. Within an e-Delphi, experts rate statements within an electronic survey, and statements meeting a minimum threshold for inclusion are taken forward [5].

### Developmental language disorder (DLD) and Phonological speech sound disorder (P-SSD)

Approximately 7% of 4 year olds present with features of DLD [6]. DLD is characterised by difficulties in understanding and using words and sentences [7]. It is closely related to, but distinct from, SSD. SSD is characterised by difficulties with producing the sounds within words. Approximately 3.4% of 4 year olds have features of SSD, with approx. 40.8% of these children having overlapping features with DLD [4]. A co-occurring profile of DLD and SSD features in the pre-school years is associated with negative long term outcomes relating to literacy [8] and persisting

communication needs [9,10], with subsequent implications for emotional well-being [1,11] and quality of life [2]. It is therefore important that this group have access to appropriate and timely intervention.

Difficulties with phonology is a shared underlying aspect of both DLD and SSD, particularly when the child's SSD features are phonological [7]. Phonology relates to the underlying sound system of a language [12], and P-SSDs occur when a child is not yet using the appropriate sound patterns for their language. In this paper (and study) we differentiate between phonology and articulation, as articulation refers to distorted sound productions (e.g., a lisp) which respond best to traditional articulation therapy rather than cognitive-linguistic approaches [13]. In contrast to children with DLD in isolation, children with P-SSDs have difficulties with their *expressive* phonology, i.e., speech production [7]. Evidence indicates that children with a co-occurring P-SSD and DLD profile have a compromised linguistic system, as evidenced by their increased speech omission patterns, in contrast to features seen within a singular P-SSD profile [14]. Recent research using network science has highlighted phonological sequence production as a unique area of difficulty in children with DLD, providing further evidence for the link between P-SSD and DLD [15,16]. Speech comprehensibility refers to listener being able to derive meaning from what a child has communicated based on their speech and wider contextual cues. This contrasts to the term speech intelligibility where meaning is more closely linked to the accuracy of the child's speech sound production [17]. The current study is concened with comprehensibility rather than intelligibility, due to the knock on impacts on the child's everyday life. For example, the adults around them not being able to understand their needs, leading to frustration [18]. Additionally, limited speech comprehensibility is further confounded for children who may additionally have limited words at their disposal – a DLD feature most frequently seen in younger children [19].

Recommended intervention approaches for children with P-SSD in isolation varies according to their error characteristics. For example, unlike children with consistant P-SSD errors, children with inconsistent P-SSD errors respond best to a core vocabulary approach [20]. This variation in response to intervention has implications for intervention development, as targeting children with both consistant and inconsonistant errors will likely impact intervention specificity [21]. This study focuses on children with consistant P-SSD errors, who make up the largest proportion of children presenting with P-SSD [13]. For young children with co-occurring DLD and P-SSD features, intervention research is in its infancy [3]. There is some indicative evidence for whole language and naturalistic approaches [22], whereby speech and language might be targeted concurrently through adult facilitation strategies [23]. However, clinical research highlights that intervention protocols are often not strictly implemented within clinical practice. Instead, speech and language therapists (SLTs) 'mix and match' different components of interventions for P-SSD or DLD to meet the unique needs of the child they are working with [24]. Reasons for this might include infeasibility of delivery in public services [25,26], which is one consequence of implementation factors, such as the views and experiences of SLTs, not being studied within intervention development processes [25]. Infeasibility within public services may relate to the aims (outcomes) of intervention research not necessarily reflecting what SLTs would prioritise within their practice [25–27], or the complex, clinical profiles of the children they support [27–28]. The active involvement of SLTs within the development of new interventions therefore has the potential to mitigate these issues [26,29].

## Complex interventions in speech and language therapy

Paediatric speech and language therapy interventions in the pre-school years are complex; they often involve multiple interacting components (e.g., multiple target areas) and levels of delivery (e.g., direct with the SLT in clinic, and indirect work through a supported parent at home) [30]. This complexity is also reflected in the three key overlapping areas of theory which might be used to underpin effective practice. For pre-school children with co-occurring P-SSD/DLD features, linguistic theories may relate to the shared underlying difficulty in phonology, and how this might be tapped into to support the development of both speech (sounds) and language (words) [31]. Recent research suggests that this shared underlying difficulty in phonology likely occurs within the context of cross-domain difficulties in sequential learning [32]. Additionally, behaviour change theory can be used to define how a SLT might a parent in carrying out activities at home;

for example, the SLT might give the parent specific feedback on how they are performing a linguistic technique with their child in clinic [33]. Explicitly specifying BCTs is important for future study replicability, as well as for our understanding of which BCTs may be 'active ingredients' within the intervention (i.e., essential to the intervention being a success) [34]. The behaviour change technique ontology (BCTO) provides a standard terminology for BCTs, for use within the development and evaluation of complex interventions [35].

A final key aspect of theory relates to implementation science, i.e., how feasible and acceptable an intervention is for SLTs applying it in everyday clinical practice [36]. Within the intervention development process, acceptability might be addressed by ensuring the key components of the intervention make sense to the SLTs who will ultimately be delivering it; this is known as 'intervention coherence' [37]. To maximise effectiveness, acceptability and implementation value, these different aspects of theory might be integrated into the development process when designing new paediatric speech and language therapy interventions.

Theory-based approaches within intervention development might be combined with other approaches (e.g., efficacy, implementation, partnership or target-population based approaches) to optimise the robustness of the intervention development process, depending on the needs of the specific clinical area being addressed [38]. For example, within paediatric speech and language therapy, a theory-based intervention development approach might be optimised by combining with a 'partnership' approach, whereby relevant professionals and people with lived experience are involved in the co-design of the intervention [39]. The relevance and 'implementation value' of newly developed interventions are likely to be increased through such co-design work within the intervention development process [30].

### Pre-study patient and public involvement (PPI) and clinician engagement (CE)

PPI and CE activities were conducted prior to this study to establish: 1. intervention outcomes of most importance for pre-school children with a co-occurring profile (aged 3:0–4:11 years), and 2. SLT perspectives on the clinical profiles of children they see in their everyday practice. PPI activities consisted of discussion and brainstorming with a 'parent panel' of three parents with lived experience within three, one hour online sessions with the lead researcher [40,41]. CE activities consisted of the lead researcher presenting on the overarching topic of co-occurring P-SSD/DLD to SLTs within three Clinical Excellence Network (CEN) meetings, after which 86 of the attending SLTs completed a poll. Both SLTs and parents highlighted a preference of aiming to expand a child's vocabulary (DLD outcome) and speech comprehensibility (SSD outcome), for children with this co-occurring profile in this age group. This aligns with research highlighting the importance of the child being able to make themselves understood by others in their everyday lives [18], and how a basic 'bank' of vocabulary, which the child is able to use functionally, is needed prior to working on more advanced areas of language, such as syntax and grammar [42].

### The current study

This study builds on this pre-study PPI and CE, as a part of a four-phase body of work aiming to develop a new, complex, theory-based intervention for pre-school children (3:0–4:11 years) with co-occurring features of DLD and P-SSD, where expressive vocabulary (DLD) and speech comprehensibility (SSD) are being targeted concurrently. The project steering group have named the intervention "SWanS" (Supporting Words and Sounds). Phases one and two included a systematic review [31] and a survey of current clinical practice [43]. The current study, phase 3, aims to bring together the wider literature and prior findings to establish consensus from SLTs on the core elements of the SWanS intervention. Key stakeholders with both professional and lived experience need to be involved throughout this process to optimise the potential relevance and applicability of these core intervention elements [30].

Research question:

**What are the agreed core elements of a novel intervention for pre-school children (3:0–4:11 years) with co-occurring features of a phonological speech sound disorder (P-SSD) and developmental language disorder (DLD) where expressive vocabulary and speech comprehensibility are joint outcomes?**

## Methods

*The process for identifying core intervention elements involved:*

1   Co-design of potential core intervention elements with the project steering group.

2   A two round modified e-Delphi with expert SLTs, to establish consensus on the core elements generated in stage 1.

Ethical approval for the publication of steering group co-design statement generation and recruitment to the subsequent e-Delphi was granted by City St George's School of Health and Psychological Sciences Research Ethics committee (ETH2223−1673 and ETH2324−2140). Consent was obtained in writing. Recruitment for the e-Delphi commenced on 17th July 2024 and completed on 29th July 2024. The ACCORD (ACcurate COnsensus Reporting Document) criteria has been used to guide our reporting for the e-Delphi [44] (S1. **ACCORD reporting guidelines**). The GRIPP (Guidance for Reporting Involvement of Patients and the Public) 2 short form has also been used to guide our reporting on the co-design work [45] (S2. **GRIPP reporting guidelines**).

This study was overseen by a Delphi working party who, in addition to the lead researcher (LR), consisted of two clinical paediatric SLTs in Professorial roles (RH and HS) and one professor of developmental psychology (NB). Critically, the Delphi working party also included members of the broader intervention development project steering group. This group consists of a parent of a child with DLD (DH), an adult with DLD (SF), three specialist SLTs (NA, EB) including one with specialist equality, diversity and inclusion (EDI) expertise (M A-K), a specialist early years teacher (LT) and a specialist bilingual educational inclusion support worker (P S-T).

### Co-design of core elements

Statements to describe potential core elements of the intervention being developed, and corresponding rationales, were derived from a systematic review [31], survey of UK clinical practice [43] and recent publications in the field of behaviour change [33,35].

In a systematic process spanning approximately 1 year, the lead researcher and steering group members of the Delphi working party co-designed statements based on these sources.

Co-design activities were based on the framework provided by Sanders and Stappers [46]; An outline of the co-design timeline is given in table 1.

In the pre-design activities (table 1), probes were used to stimulate discussion and lay the groundwork for later idea (i.e., statement) generation. Key probes included findings from a systematic review and survey of clinical practice, which were used as a basis for brainstorming ideas together on online interactive whiteboards [47].

The second part of the co-design process focused on 'generation', whereby group members worked together on interactive whiteboards [47] to generate statements to describe core intervention elements, based on their prior discussions. Individual meetings involved the identification of potential Behaviour Change Techniques (BCTs) for the intervention being developed, based on prior work [33]. These BCTs were mapped onto the updated behaviour change technique ontology (BCTO) [35] by the lead researcher prior to discussions with steering group members.

### Two round modified e-Delphi

Co-design activities resulted in 47 statements to describe potential core intervention elements, in S6. **Round 1 questionnaire (47 statements).**

A modified e-Delphi methodology was selected to establish consensus by expert SLTs on the core intervention elements generated by the project steering group. An e-Delphi methodology was selected as it is a structured way of gaining consensus across a group of panellists, whilst minimising respondent bias by ensuring anonymity [5]. The Delphi was conducted electronically (an e-Delphi), to enable responses to be given at a time most convenient to panellists and thus facilitate uptake [48].

**Table 1. Timeline for the co-design of core intervention elements.**

| Date *(co-design phase)* | Activity | Output/example |
|---|---|---|
| **June 2023** *(pre-design)* | Sub-group* brainstorming- implications of the phase 1 (systematic review) findings. | Online whiteboard with implications of the systematic review findings, discussion probe for comparing systematic review and survey findings, both of which are included in S3. **Pre-design steering group activities** |
| **October/November 2023** *(pre-design)* | 1:1 discussion with each group member- opportunity to add/expand on the Online whiteboard. | |
| **October/November 2023** *(pre-design)* | 1:1 discussion with each group member- clinical practice survey findings [43] and similarities and differences with the systematic review findings. | |
| **December 2023** *(generative)* | Sub-group brainstorming regarding potential intervention characteristics arising from the systematic review and survey findings and prior discussions. | Online whiteboards of possible intervention characteristics and steering group ideas, in S4. **Generative co-design activities** |
| **March 2024** *(generative)* | 1:1 discussion with each group member- opportunity to add/expand on the online whiteboard. | |
| **May 2024** *(generative)* | 1:1 discussion with each group member, each member reviewing key behaviour change techniques (BCTs) leading to generation of e-Delphi statements, in S5. **BCTs mapped onto the behaviour change technique ontology (BCTO).** | Completed list of e-Delphi statements ready for round 1 of the e-Delphi, in S6. **Round 1 questionnaire (47 statements).** |

*Allocation to sub-groups varied between meetings. However, to ensure a spread of expertise, sub-groups were devised so that they each contained a person with lived experience (DH or SF) and a professional with equality, diversity and inclusion (EDI) expertise (M A-K or P S-T) where possible.*

**E-Delphi panellists.** For the e-Delphi, we defined experts as UK Health and Care Professions Council (HCPC) registered SLTs with a minimum of 5 years relevant clinical experience, and a self-identified specific interest in intervention for pre-schoolers with co-occurring SSD and DLD features. Panellists also had to state that they attempt to keep up to date with the evidence base. As the lead researcher, LR was responsible for panellist selection.

Justification of our criteria for 'expert panellist' is twofold. Firstly, clinical experts can contribute to the refining of statements, based on their clinical expertise and insight. Secondly, SLTs will be carrying out the intervention with families. To optimise 'implementation potential', it is therefore essential that core elements and underlying rationales are coherent to them [37].

**Panel size.** A panel of over 30 may not necessarily improve the quality of results within a homogeneous e-Delphi [49,50]. As attrition can be particularly high within an e-Delphi process [51], the researchers sought to recruit 60 panellists to allow up to 50% attrition in the final round.

**Recruitment.** The research team used their national networks and social media to advertise the study. This included emails via relevant CENs, such as the London SSD CEN and Communication in the Early Years CEN. Additionally, SLTs who previously signed up to intervention development project mailing list were notified of the recruitment for the e-Delphi via email. A link to register interest in taking part in the e-Delphi, including the consent, demographic and participant information sheets, was embedded into emails and social media posts. A total of 50 potential panellists registered their interest.

**Round 1 questionnaire and piloting.** The questionnaire for round 1 included the 47 co-designed intervention elements, together with corresponding rationales, in S6. **Round 1 questionnaire (47 statements).** The questionnaire was separated into 3 key sections: target setting (11 statements), intervention content (20 statements), intervention delivery (8 statements). Following this were 4 statements regarding 'manual contents' (i.e., what resources the panellists would like the intervention manual to include). The questionnaire concluded with 4 statements regarding potential future adaptations to the intervention. This questionnaire was created using Qualtrics Software© [52], and initially trialled for functionality with

three SLTs. Amendments were made as needed; for example, re-ordering one of the statements so that the questionnaire followed a logical progression.

**Pre-specified criteria *for* consensus.** In accordance with Delphi studies of a similar nature (i.e., a homogenous panel, aiming to achieve consensus on a specific issue relating to clinical practice), consensus was pre-defined as over 75% of responses (from a minimum of 30 panellists) rating the statement as either appropriate/highly appropriate [7]. To ensure a minimum level of consistency the interquartile range also had to be 1 or less, with this being identified as a suitable consensus indicator for 5 point scales [53].

As in line with our pre-defined process, statements not achieving consensus in round 1 but still rated as appropriate/ very appropriate by over 50% of panellists were revised by the project steering group, reworded in response to free text comments, and presented again for rating in round 2. Statements not achieving consensus in round 1 and having 50% or below ratings as appropriate/very appropriate were discarded. Statements not achieving consensus in round 2, but still achieving 50% or above of appropriate/very appropriate ratings, were not defined as reaching consensus but still elaborated on in the discussion of the findings. There was one amendment to this process in regard to four statements which reached consensus in round 1 but required merging; this is elaborated on in the results.

**Round one.** On 5/8/2024, a personal link to the first round of the e-Delphi was sent to all individuals who registered their interest via Qualtrics. Panellists were asked to rate questionnaire statements (i.e., intervention elements) and corresponding rationales on a 5-point Likert scale from highly inappropriate (1) to highly appropriate (5). Alongside each statement was a free text box where panellists had the option to elaborate on their response. This elaboration was explicitly recommended where a rating of 3 or less was given, to enable the research team to understand how statements not achieving consensus could be amended. The research team decided not to make open text comments mandatory, due to the potential time burden for panellists.

Panellists were initially given two weeks to complete the form, with automatic email reminders to complete it one week, 48 hours, and 24 hours before the deadline. The deadline was then extended by one week to try and capture more panellists. The piloting process indicated that it would take between 45 minutes and an hour to complete the survey. Although free text comments were optional, 86% (31/36) of panellists left a free text comment for one or more statement.

**Interim feedback.** Two weeks after the completion of round 1, panellists were emailed a summary of their responses. The feedback consisted of both the percentage of appropriate/very appropriate responses and medians and inter-quartile ranges from the full dataset, with these calculations being more robust to the effects of outliers [54].

**Round two.** On 19/9/2024, a personal link to the round 2 questionnaire was emailed to each of the 36 panellists who completed round 1. The results of round 1 informed round 2 questionnaire content in the following key ways: the second questionnaire included re-worded statements that did not achieve consensus in round 1 and two merged statements (relating to target setting) which arose from round 1 responses. Round 2 also included the 2 reworded statements which panellists stated that they did not fully understand.

Again, there were automated reminders to complete the form, and the deadline was also extended by a week to capture as many panellists as possible.

## Results

An overview of the e-Delphi process and results are in fig 1.

### Panellists

The 36 panellists (from 50 invites) were from 11 different regions across the UK in round 1 (table 1). All were female. Main employer was the NHS (UK National Health Service) for 30/36 panellists, independent practice for 5/36 panellists, and educaton for 1/36 panellists. The median number of years qualified was 16–20 years (table 2).

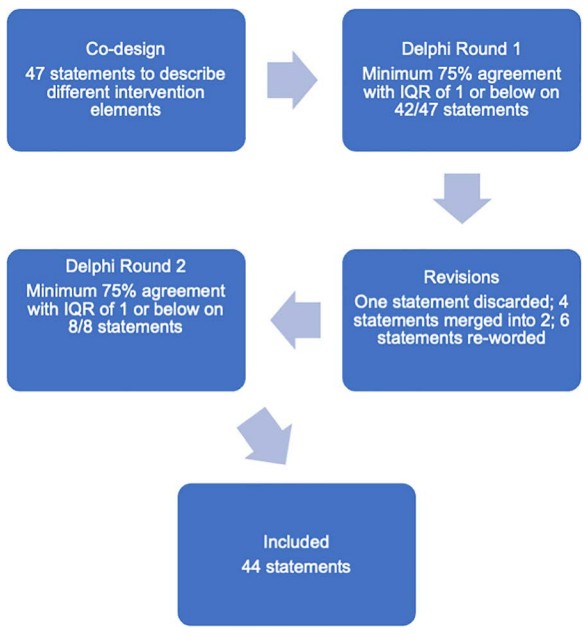

**Fig 1. Flow chart of the e-Delphi process.**

**Table 2. Panellist location and years of experience.**

| Region | No. of panellists (%) |
|---|---|
| East Midlands | 1 (2.7%) |
| East of England | 3 (8.3%) |
| London | 5 (13.8%) |
| North East England | 5 (13.8%) |
| North West England | 2 (5.5%) |
| Scotland | 2 (5.5%) |
| South East England | 5 (13.8%) |
| South West England | 5 (13.8%) |
| Wales | 2 (5.5%) |
| West Midlands | 5 (13.8%) |
| Yorkshire and Humber | 1 (2.7%) |
| **Years of experience** | **No. of panellists (%)** |
| 5-10 | 7 (19.4%) |
| 11-15 | 10 (27.7%) |
| 16-20 | 5 (13.8%) |
| 21+ | 14 (38.8%) |

## Round 1

In round 1, 42/47 (89.3%) statements reached consensus, with percentage of agreement ranging from 77.8% to 100%. Table 3 shows items reaching consensus in round 1, including % percentage agreement, median and inter-quartile ranges. The table presents the statements in order of percentage agreement within the 5 sections of the questionnaire;

**Table 3. Statements reaching consensus in round 1.**

| Statement | % (n/n) | Median and inter-quartile range |
|---|---|---|
| *Intervention targets (11 statements)* | | |
| Statement 7. Agree outcome goal BCT:<br>The parent and clinician will agree the child's targets.<br>*Rationale: By agreeing the child's targets, parents have more agency, and therefore will potentially have more motivation to carry out the targets at home. The targets are also more likely to be relevant to the child.* | 100% (36/36) | 5(5/5) |
| Statement 1. The intervention will include a vocabulary target which is based on both developmental norms and words the child is most likely to need within daily life.<br>*Rationale: This will facilitate a short-term functional impact for the child as well as aiding future language growth/sentence construction.* | 97.2% (35/36) | 5 (4/5) |
| Statement 2. The intervention will include a phonological awareness target which is suitable for their developmental level (i.e., attention and listening capacity).<br>*Rationale: Phonology is a shared difficulty in pre-schoolers with this profile. Working on phonology has the potential to impact speech production and lays the groundwork for later literacy development.* | 97.2% (35/36) | 5 (4/5) |
| Statement 3. The intervention will include a sound awareness target, based on the child's speech process errors.<br>*Rationale: Consistent speech process errors are mislearnt sound patterns. By targeting a process, this mislearning can be challenged. It also lays the foundation for future direct work for speech if needed.* | 97.2% (35/36) | 5 (4/5) |
| Statement 6. Goal strategizing BCT: The parent and clinician will talk through barriers to implementing techniques and review strategies for overcoming these barriers.<br>*Rationale: By making potential barriers explicit, the SLT can problem solve with the parent how to over-come them, so that implementation is as accessible for the parent as possible.* | 97.2% (35/36) | 5(4/5) |
| Statement 8. Action planning BCT: The parent and clinician will plan when, where and how the intervention techniques will be carried out.<br>*Rationale: By having a clear, joint plan for implementation, implementation at home will be more feasible and accessible.* | 97.2% (35/36) | 5(4/5) |
| Statement 11. Intervention techniques and activities will be explicitly linked to the target they are addressing.<br>*Rationale: By having a clear link between targets and content, parents and wider support networks will better understand the purpose of what they are doing.* | 97.2% (35/36) | 5(4/5) |
| Statement 10. Before setting targets, the clinician will liaise with other services (where relevant) to ascertain prior and current support received.<br>*Rationale: By liaising with other services, the SLT can start "where the family is at" and make sure that the family is getting wider support they might be entitled to.* | 94.4% (34/36) | 4(4/5) |
| Statement 5. Set behaviour goal BCT (behaviour change technique): The parent and clinician will set goals for how the parent will carry out the intervention techniques.<br>*Rationale: By doing this jointly, the SLT can make sure that the way in which the parent carries out the intervention techniques is both relevant and feasible for them.* | 91.7% (33/36) | 5(4/5) |
| Statement 4. The intervention will include a comprehensibility target (e.g., use of aided language boards, telling the child "show me"), based on which activities their comprehensibility is most impacting.<br>*Rationale: This will have an immediate positive impact on the child's functional communication in everyday life, whilst their speech intelligibility is still developing.* | 88.8% (32/36) | 4(4/5) |
| Statement 9. Where the child is bi/multilingual, intervention targets will align with any available norms for their home language.<br>*Rationale: Different languages have different phonological inventories (sound patterns). It would be inappropriate to target an error process in the child's home language if this does not typically develop until they are much older.* | 88.8% (32/36) | 5(4/5) |
| *Intervention content (19 statements)* | | |
| Statement 14. Sound awareness relating to error processes will be targeted through hybrid use of focused auditory stimulation (also known as auditory bombardment), recasting, visual cues (e.g., cued articulation), and exposure to word contrasts.<br>*Rationale: Helpful techniques for developing sound awareness (of an error process) are inter-connected and can complement each other. Visual referents can reinforce the auditory processing of sounds.* | 100% (36/36) | 5(4/5) |

*(Continued)*

 

**Table 3.** (Continued)

| Statement | % (n/n) | Median and inter-quartile range |
|---|---|---|
| Stataement 17. Knowledge development behaviour BCT: The Clinician will explain to the parent the rationale for intervention and what speech and language therapy is.<br>*Rationale: By giving parents this knowledge, this will empower them to more fully understand their child's needs and make informed choices.* | 100% (36/36) | 5(5/5) |
| Statement 19. Provide feedback on outcome of behaviour BCT: The clinician will give feedback to the parent on the impact of them conducting the intervention techniques (i.e., the change observed in the child).<br>*Rationale: By drawing attention to the positive consequences on the child of the parent carrying out the techniques, parents will feel more encouraged and this could support motivation.* | 100% (36/36) | 5(4/5) |
| Statement 25. Demonstrate the behaviour BCT: The clinician will model techniques for the parent to see.<br>*Rationale: The parent is more likely to learn the technique if given a tangible example in person.* | 100% (36/36) | 5(5/5) |
| Statement 27. Reduce cue frequency BCT: The clinician will gradually withdraw prompting/cues when the parent is carrying out the technique.<br>*Rationale: Withdrawing cues will enable the parent to gradually become more independent when carrying out the intervention techniques.* | 100% (36/36) | 5(4/5) |
| Skill development behaviour BCT: The parent will practice carrying out the technique with the clinician. (27)<br>*Rationale: The parent is more likely to learn the technique if they get hands on practice with a clinician guiding them.* | 100% (36/36) | 5(4/5) |
| Statement 12. Vocabulary will be targeted through adult exposure according to the child's level of language development (e.g., single word modelling for minimally verbal children, adding a word for early combiners).<br>*Rationale: Exposure in a variety of contexts and structures is in line with cross-situational learning principles for vocabulary development.* | 97.2% (35/36) | 5(4/5) |
| Statement 16. Strategies to support with speech comprehensibility in everyday life (e.g., selecting vocabulary for picture boards) will be co-produced with the child's family.<br>*Rationale: By developing these strategies (and aids for delivering them) together, the SLT is better able to make sure that they are meaningful to the child and family and are culturally inclusive.* | 97.2% (35/36) | 5(5/5) |
| Statement 28. Provide positive social consequence for the behaviour BCT: The clinician will provide praise when the parent is making progress with implementing the techniques.<br>*Rationale: As well as highlighting to the parent what they should continue doing, this will make the experience more positive for the parent and support motivation.* | 97.2% (35/36) | 5(4/5) |
| Statement 31. The intervention will include a flexible option of activities/routines for parents to incorporate relevant language techniques into, with support to identify their own.<br>*Rationale: By enabling parents to choose the best activities/routines to integrate the language techniques into, they are more likely to be culturally inclusive, reflect the child's current preferences, and feasible for implementation at home.* | 97.2% (35/36) | 5(4/5) |
| Statement 13. Phonological awareness will be targeted through syllable segmentation activities, or word segmentation if the child is not ready for syllable work yet.<br>*Rationale: Poor syllable segmentation, and persistent difficulties with polysyllabic words, is a key feature for children with this profile. These sentence and word level skills are foundational to future literacy development and lay the groundwork for more advanced phonological awareness work in the future.* | 94.4% (34/36) | 5(4/5) |
| Statement 15. Speech comprehensibility will be targeted through integration of strategies into everyday activities- e.g., using aided language boards at home, telling the child "show me".<br>*Rationale: Such integration is more likely to have an immediate, positive impact for the child. By focusing on the child's everyday activities, these strategies are more likely to be relevant to the individual child and family.* | 94.4% (34/36) | 5(4/5) |
| Statement 18. Provide feedback BCT: The clinician will give feedback to the parent about how they are conducting intervention techniques.<br>*Rationale: Feedback will help the parent to maintain and improve on their technique implementation.* | 94.4% (34/36) | 5(4/5) |

*(Continued)*

**Table 3.** (Continued)

| Statement | % (n/n) | Median and inter-quartile range |
|---|---|---|
| Statement 20. Self-monitor behaviour BCT: The parent will monitor how they are continuing with techniques/activities at home as an informal measure of progress.<br>*Rationale: By monitoring how they are getting on, the parent will be able to modify their input accordingly. This will also help them to maintain independence with technique use between sessions with the Speech and Language Therapist (SLT).* | 94.4% (34/36) | (4/5) |
| Statement 22. Social Support BCT: The clinician will take the time with the parent to establish who is best placed to deliver the intervention techniques, and wider family support which might facilitate intervention implementation.<br>*Rationale: Wider family networks and friends may be able to facilitate the carrying out of techniques and provide support for the parent at home.* | 91.7% (33/36) | 5(4/5) |
| Statement 30. 'Readiness' for direct elicitation of speech/language will be jointly decided by the clinician and family (and child where possible).<br>*Rationale: The parent knows the child best and can give insight into how the child has responded to direct elicitation in the past.* | 91.7% (33/36) | 5(4/5) |
| Statement 21. Self-monitor outcome of behaviour BCT: The parent will monitor the impact of their work with their child at home as an informal measure of progress.<br>*Rationale: By seeing potential positive impacts of them delivering intervention techniques with their child, parents may feel more encouraged and motivated. Additionally, this will empower them to monitor their child's speech and language development when their child is not receiving therapy.* | 88.9% (32/36) | 4(4/5) |
| Statement 29. Techniques will be primarily input based, with flexibility to elicit speech/language directly from the child if they demonstrate readiness for this.<br>*Rationale: The child may lack confidence or have difficulties sustaining their attention. Heavily encouraging the child to produce speech/language if they are not ready may have negative consequences for their future involvement in speech and language therapy.* | 80.6% (29/36) | 4.5(4/5) |
| Statement 23. Inform about social consequences BCT: The clinician will talk to the parent about the potential social consequences (positive or negative) of carrying out the intervention techniques. Note: positives emphasised, negatives kept to a minimum and to be discussed sensitively.<br>*Rationale: The parent understands why the intervention and related techniques are important. This may support motivation.* | 77.8% (28/36) | 4 (4/5) |
| Intervention delivery (6 statements) | | |
| Statement 33. The language aspects of the intervention will primarily be delivered through the supported parent using language facilitation techniques in the child's everyday life.<br>*Rationale: Multiple exposures to language in meaningful contexts for the child will facilitate child engagement and wider generalisation of language learnt. The parent knows their child best and can implement techniques in multiple relevant contexts as they see fit (e.g., bathtime, at the park, on holiday, visiting the dentist).* | 100% (36/36) | 5(4/5) |
| Statement 39. Generalisation in learning BCT: Parents will be asked to deliver the intervention techniques from clinic at home.<br>*Rationale: By generalising their use of language techniques at home, the child will be supported in developing their language within meaningful environments.* | 100% (36/36) | 5(4.25/5) |
| Statement 35. Guidance will be given regarding an intervention duration (including number of sessions and spacing of sessions).<br>*Rationale: Having a duration for the intervention will enable the SLT to plan accordingly and help the parent to know what to expect from the outset.* | 97.2% (34/35) | 5(4/5) |
| Statement 37. Intervention activities should be changed if the child lacks motivation or is not enjoying them.<br>*Rationale: If the child is not enjoying the activity they are more likely to lose attention; this will have a knock-on effect on their subsequent language learning.* | 97.2% (35/36) | 5(4/5) |
| Statement 34. Guidance will be given regarding dosage ranges for intervention techniques (technique dosage per activity).<br>*Rationale: Having an approximate idea of how many times a technique should be used in an activity will enable both the parent and SLT to know what they are aiming for, and what is most likely to be effective.* | 94.4% (34/36) | 5(4/5) |

*(Continued)*

**Table 3.** (Continued)

| Statement | % (n/n) | Median and inter-quartile range |
|---|---|---|
| Statement 38. Add objects to the environment BCT:<br>Parents to bring the child's favourite toys/books/items from home into the clinic.<br>*Rationale: By having toys/objects from home brought in the child may feel more comfortable. It would also allow the SLT to use them when demonstrating language techniques during the session.* | 83.3% (30/36) | 5(4/5) |
| Intervention manual contents (4 statements) | | |
| Statement 42. The intervention manual will include targets which can be selected across the 4 target areas, as well as guidance around progression and links to relevant non-English speech/language norm resources.<br>*Rationale: Targets chosen from the manual can be explicitly linked to the intervention content. Children often progress at different rates. Guidance around progression will support the SLT in deciding where to go next if the child has made a lot of progress in relatively few sessions (or conversely, very little progress).* | 100% (36/36) | 5(4/5) |
| Statement 43. The intervention manual will include key handouts for parents, which can be personalised.<br>*Rationale: Having something to take away from the session will support the parent in remembering what was covered. By being able to personalise the handouts, they will be more relevant to that specific child and family.* | 100% (36/36) | 5(5/5) |
| Statement 40. The intervention manual will include a flexible discussion guide to support both the clinician in getting to know the family, and the family in understanding more about speech and language therapy.<br>*Rationale: Understanding the child and their family is the foundation for what comes next. Important for the SLT to develop a positive rapport with the parent from the outset and have an understanding of what barriers (and supports) they have in everyday life.* | 94.4% (34/36) | 4(4/5) |
| Statement 41. The intervention manual will include guidance on what 'readiness' for direct speech work (i.e., eliciting words from the child) could look like.<br>*Rationale: Definition of 'readiness', and what this looks like in practice, may vary (e.g., is it to do with confidence, attention, or both?).* | 94.4% (34/36) | 5(4/5) |
| Potential (future) adaptations (2 statements) | | |
| Statement 44. Option for the intervention to be delivered with education professionals, in educational settings. (potential future adaptation) | 94.5% (34/36) | 5(4/5) |
| Statement 47. Future development of a shared 'app' for parents and clinicians as a tool for guiding intervention delivery and monitoring progress. (potential future adaptation) | 94.5% (34/36) | 4(4/5) |

intervention targets (11 statements), intervention content (19 statements), intervention delivery (6 statements), manual contents (4 statements), future adaptations (2 statements).

## Review process and rewording

De-identified raw data from round 1 was shared with the Delphi working party. The lead researcher liaised with the project steering group regarding statements which required rewording. How the statements were revised is discussed in more detail below.

**Discarded and re-worded statements.** One statement was discarded ahead of round 2 (24. "inform about environmental consequences BCT"). Although it reached the minimum threshold for re-wording in round 2, free text comments indicated uncertainty about what the statement meant. Following a re-review of the changes to the BCTv1 and BCTO criteria, it was confirmed that this item was only relevant when combined with statement 23 (inform about social consequences BCT) in the original BCTv1 taxonomy, and therefore irrelevant when using the updated BCTO.

Six statements were reworded for round two. The original and reworded statements are given in S7 **Reworded statements**. Examples were added in two of the statements because a panel member commented that they did not fully understand them. The other statements were reworded because they failed to achieve consensus.

Four sets of statements that were merged into two are also presented in S7 **Reworded statements**. They were refined in response to panellist concerns about the number of target areas, and uncertainty about the overlap between 'speech awareness' and 'phonological awareness'. Full details of the round 1 responses which prompted the refinement of these targets are provided in the S8. **Interim feedback document**. The two (initially separate) targets relating to sound and phonological awareness were reduced to a single target, together with a corresponding statement about activities. The Pascoe, Stackhouse and Wells' speech processing model [55] was used to inform the wording for two statements relating to the amended target area.

### Round 2

Of the panellists who completed round 1, 97.2% (35/36) returned for round 2, which contained 8 statements. All revised statements achieved consensus at round 2, with percentage of agreement ranging from 77.2% to 100%. Detail is given in table 4 regarding percentage agreement, median and interquartile range, and change in percentage agreement across the 2 rounds (where applicable).

Following the second round, consensus had been reached on 44 (100%) intervention elements (statements) in total; 10 core elements relating to intervention targets (22.7%), 18 core elements relating to intervention content (40.9%), 8 core elements relating to intervention delivery (18.2%), 4 core elements relating to the intervention manual (9.1%), and 4 core elements regarding future adaptions (9.1%).

## Discussion

Consensus has been reached on the core elements of a novel intervention for pre-school children with co-occurring P-SSD/DLD features, where expressive vocabulary and speech comprehensibility are joint outcomes of interest. Thirty-six of these agreed core elements relate to intervention targets, content, and delivery; 4 refer to manual contents; 4 refer to potential future iterations. Consensus was achieved by combining a structured, multi-stakeholder co-design process with a subsequent two round e-Delphi with 35 expert SLTs. The core elements can now be integrated into a new intervention for children with a co-occurring P-SSD/DLD profile (SWanS- "Supporting Words and Sounds"), who are highly vulnerable but under-served within research. Below we discuss key over-arching findings and implications for intervention development.

### Intervention flexibility

Consensus was reached on an intervention with 3 target areas (vocabulary, psycholinguistic, speech comprehensibility), where the SLT carries out activities with the child in clinic whilst using behaviour change techniques (BCTs) to support the parent with carryover activities at home. One key element of contention was whether the parent would carry out speech/psycholinguistic activities at home, in addition to vocabulary. Research highlights the value of parents carrying out such language activities, as they can expose the child to vocabulary in real world, meaningful contexts [56]. However, the relative benefits of the parent carrying out speech related activities, versus the clinician, is less clear. Parents carrying out speech work at home can increase dosage, and therefore enhance potential intervention effectiveness in some children [57]. However, there is also a risk of over-burdening the parent [58]. The panellist responses reflected this tension. Round one showed that some panellists were strongly against parents not doing speech related activities at home, whereas other panellists strongly agreed that it should be the role of the SLT to provide this support in clinic. Consensus was reached when this element was made flexible, with the option for tailoring according to feasibility for parents. Flexibility relating to deliverer (e.g., by the SLT, parent, or both) is not routine across interventions in the paediatric speech and language therapy field [31]. However, by integrating flexibility into the intervention design, it is more likely to meet the needs of different families who are seen within day to day clinical services [27,59,60].

Findings also highlight the complexity of balancing theory and 'implementation potential' within the intervention development process, where partnership and theory based approaches have equal weighting. This complexity was reflected when assimilating targets 2 and 3 (sound and phonological awareness) into a single (psycholinguistic) target area. The new

**Table 4. Reworded statements reaching consensus in round 2.**

| Statement (questionnaire item no.) | Round 2% (n/n) *Round 1% (n/n)* | Round 2 median and inter-quartile range *Round 2 median and inter-quartile range* | Change in % agreement from round 1 (where applicable) |
|---|---|---|---|
| Intervention techniques and activities will be explicitly linked to the target they are addressing. For example, explicitly stating that language modelling is to help the child achieve their language target. (7) *Rationale: By having a clear link between targets and intervention content, parents and wider support networks will better understand the purpose of what they are doing.* | 100% (35/35) *97.2% (35/36)* | 5(4/5) *5(4/5)* | ↑ 2.8% |
| Inform about social consequences BCT (behaviour change technique): The clinician will talk to the parent about the potential social consequences (positive or negative*) of carrying out the intervention techniques. For example, that an increase in language (as a result of carrying out a technique such as modelling) could help them to interact with their peers with more ease. (8) *positives emphasised, negatives kept to a minimum and to be discussed sensitively *Rationale: This will help the parent to understand why the intervention and carrying out the techniques are important. This may also support motivation.* | 97.1% (34/35) *77.8% (28/36)* | 4(4/5) *4(4/5)* | ↑ 19.3% |
| If the intervention is being delivered through an interpreter, double time should be allocated (this time might be spread across sessions, rather than the child having to attend one long session). (4) *Rationale: This is in accordance with RCSLT guidelines. Sessions delivered through interpreters are more time consuming. Additionally, the SLT may need to brief the interpreter about what is expected of them prior to the session.* | 91.4% (32/35) *75% (27/36)* | 5(4/5) *5(3.25/5)* | ↑ 16.4% |
| Depending on the child's developmental level, their speech sound system (i.e., psycholinguistic profile) will be supported through syllable/phoneme level awareness activities. These activities might integrate exposure to sound/word contrasts using techniques such as auditory stimulation (also known as auditory bombardment), recasting and visual cues (e.g., cued articulation). (2) *Rationale: These activities address underlying difficulties in motor programming and/ or phonological representations, which often occur in children with this profile. They also support vocabulary development.* | 88.6% (31/35) *N/A* | 5(4/5) *N/A* | N/A |
| In the future, there would be an option for the intervention to be delivered in a hybrid face-to-face and online format. Online sessions might focus on parent coaching, with face-to-face sessions consisting of direct work with the child. (5) *Rationale: Having this option could increase the accessibility of the intervention for families who may find it difficult to attend clinic on a regular basis. The clinician will also be able to "see" the parent/child in their home environment, without the additional travel time involved when conducting a home visit.* | 88.6% (31/35) *72.2% (26/36)* | 4(4/5) *4(3/4)* | ↑ 16.4% |
| The intervention will include a psycholinguistic target based on the child's developmental level (including attention and listening and phonological awareness ability), and speech error characteristics. (1) *Rationale: Children with this co-occurring profile are likely to have underlying difficulties with their motor programming and/or phonological representations, which influences their speech output.* | 85.7% (30/35) *N/A* | 5(4/5) *N/A* | N/A |
| The speech/ psycholinguistic aspects of the intervention will be primarily managed and delivered by the clinician in clinic, with the flexibility for parents to carry out simple support strategies at home if feasible (e.g., being taught a small number of cued articulation signs to use whilst language modelling). (3) *Rationale: It may not be feasible for some parents to do carry over work on all aspects of the child's intervention at home. By primarily working on the more complex aspects of speech in clinic, the SLT is able to control for exposure to items with particular sounds and use their specialist expertise to modify their input in response to the child's productions.* | 77.2% (27/35) *66.7% (24/36)* | 4(4/5) *4(3/5)* | ↑ 10.5% |
| In the future, there would be an option for the intervention to be delivered via the clinician doing home visits (with the parent conducting carryover activities between visits). (6) *Rationale: Having this option could increase the accessibility of the intervention for families who may find it difficult to attend clinic and have limited access to technology.* | 77.2% (27/35) *66.7% (24/36)* | 4(4/5) *4(3/5)* | ↑ 10.5% |

target area needed to be relevant to everyday practice but also informed by theory. The challenge posed to the research team was that children with a consistent P-SSD and co-occurring DLD features remain a heterogeneous group. For example, their speech errors may be typical or atypical [61]. However, although having overarching participant characteristics is essential, being highly granular within the intervention development process runs the risk of limiting the potential of the intervention before it has been trialled [62]. Therefore, the updated (psycholinguistic) target area and option of corresponding intervention techniques (e.g., syllable segmentation, speech recasting) have been kept deliberately flexible. This will enable researchers to investigate the responses of different children to the intervention when carrying out further proof of concept work in the initial trailing phase of the intervention development-evaluation process [30]. The target and corresponding techniques can then be further refined, if needed. This is not uncommon within complex intervention research, where the development-evaluation process is rarely viewed as a linear process [62].

## Service delivery implications

There is no guarantee that the consensus of re-worded statements in round 2 was a direct result of panellists disregarding their service limitations in their responses. However, free text comments in round 1 highlighted a clear discrepancy between how panellists might ideally carry out their practice and how this is actualised within the context of service limitations. This is a well-known friction, as highlighted in other studies of clinical practice in paediatric speech and language therapy [27,60]. With careful and strategic planning, tailoring interventions can allow healthcare professionals to meet the needs of their local population, in a way which is effective but minimises public cost [63]. However, our findings highlighted that SLTs may not have the required resources within their service to provide an equitable service, such as allocation of double time when delivering a session with an interpreter; this conflicts with professional guidelines [64]. This could potentially widen pre-existing healthcare inequalities for families who often already face marginalisation [65,66], and hinder effectiveness of speech and language therapy intervention for non-English speaking children [67]. For new interventions being developed, it is therefore imperative that allocated time to work with an interpreter, in line with institutional guidelines, is an explicit component. The fact that consensus was achieved in round 2 on this double time allocation indicates that this is a preference of SLTs, regardless of the limitations of their service structure.

Variation and service structure barriers were also reflected in the consensus rates for statements relating to 'potential future adaptations'. These are *potential* future adaptations which will take place when/if there is evidence to support the effectiveness of the intervention in its current form (i.e., face-to-face sessions with a SLT in clinic and carryover work with a supported parent at home). In round 2, these potential future aspects of delivery (such as home visits or hybrid in person/online) all achieved consensus when panellists were asked to base their response on what they would do rather than on what they can currently provide. However, barriers within service delivery are a reality which will need to be addressed, with the balancing of stakeholder preferences and technical/political concerns being key when addressing public value [68]. Therefore, prioritisation work will need to take place in advance of such adaptation work to decide which adaptations are both preferred by stakeholders and best fit the contexts for which they are being developed [57].

## Wider relevence

This e-Delphi process, and subsequent intervention, is for pre-school children with co-occurring DLD and consistant P-SSD features. We caution against the overgeneralisation of our findings, particularly in regard to the statements relating to intervention targets and corresponding activities. However, some of the core elements idenitfied may be relevent to interventions for other SSD sub-types and/or DLD interventions which target different outcomes. This is because supporting parents to continue with activities at home is a feature of many pre-school speech and language therapy interventions [69,70], across diagnoses. Researchers may find it helpful to draw upon our statements, particularly those relating to BCTs, when developing interventions in the future.

## Strengths and limitations

A strength of this study included its two-step approach: co-design of potential intervention elements with key stakeholders, followed by the 2-round modified e-delphi. By using a structured, in depth approach to statement generation [46], our co-design approach moved beyond consultation to co-learning and collective action [71]. This co-design process was intensive and started over a year prior to formal e-Delphi statement generation, when the project steering group were involved in conducting and interpreting the supporting systematic review [31]. We would argue that this time was a worthwhile investment, as evidenced by the 89.3% of statements that achieved consensus in the first e-Delphi round. This high level of consensus is likely, in part, due to the co-design approach to statement generation increasing their relevance. Critically, the co-design approach also enabled us to generate statements which not only reflect the best available evidence but centre the voices of a diverse range of key stakeholders. Ultimately, this should enhance the implementation value of the intervention, and its meaningfulness to SLTs and families [30].

One potential limitation of the e-Delphi was the representation of panellists and resulting limitations to statistical analysis. A recruitment target of over 30 panellists was aimed for prior to the study commencing, and this was achieved. However, this was split between those who worked primarily for the NHS (30 panellists in both rounds) and those who worked in independent practice/education (6 panellists in round 1, 5 panellists in round 2). Given the service delivery implications of our findings, it may have been illuminating to run statistical tests to compare responses between NHS and non-NHS panellists. We recommend this as a consideration to others developing Delphi protocols which may include both NHS and non-NHS practitioners.

A further potential limitation of the study is that although core elements have been identified, they will likely need further refinement prior to operationalising within a trial. We have statements to describe core elements; however, free text comments revealed further nuances which can be explored in subsequent intervention development work. Such iterative work is typical within the intervention development-evaluation process, as interventions grow and adapt in response to new information [64]. Related to this is the need to carry out further work with parents on how to best optimise the core intervention elements. Although a diverse range of stakeholders were involved in the statement co-design process, further involvement of parents when refining the core intervention elements could further increase the acceptability of the intervention to parents within clinical practice.

## Conclusion

Following a multi stakeholder co-design process and subsequent two round e-Delphi, we have successfully achieved consensus on the core elements of a new intervention for pre-school children with co-occurring DLD/P-SSD features. These elements relate to 3 key target areas: expressive vocabulary, psycholinguistic/speech processing, and speech comprehensibility. The target areas are to be addressed through direct work with a SLT in clinic, who will use the identified BCTs to support the parent with some delivery at home. To facilitate applicability within everyday practice and to meet the needs of individual families, flexibility is needed when selecting these home activities and intervention techniques. To explore their effectiveness in practice, the core intervention elements require testing and may undergo further refinement in response to findings from initial intervention trials.

## Supporting information

**S1. ACCORD reporting guidelines.**
(DOCX)

**S2. GRIPP reporting guidelines.**
(DOCX)

**S3. Pre-design steering group activities.**
(DOCX)

**S4. Generative co-design activities.**
(DOCX)

**S5. BCTs mapped onto the behaviour change technique ontology (BCTO).**
(DOCX)

**S6. Round 1 questionnaire (47 statements).**
(DOCX)

**S7. Reworded statements.**
(DOCX)

**S8. Interim feedback document.**
(DOCX)

## Acknowledgments

The authorship team would like to thank members of the e-Delphi panel, including: Bethan Taylor, Natalie Bagwell, Katy Orr, Mary Quinn, Claire Pimenta, Claire Butler, Jeni Halls, Shula Burrows, Charlie Ayling, Caroline Rendle, Jan Broomfield, Amelia Louth, Claire Torrance, Sandy Chappell, Katie Prieto, Sue Maughan, Katy Liriano, Josie Haggar, Jane Taylor, Stephanie Dunstan, Marie Newton, Rebecca Hammonds, Vicky Baxter, Kirsty English, Anna Benedict, Alison George, Anne Dodd, Elise Lightbody, Catherine Allison, Katie Anthony, Annie Teather

## Author contributions

**Conceptualization:** Lucy Rodgers, Nicola Botting, Natalie Abdo, Ros Herman.

**Data curation:** Lucy Rodgers, Natalie Abdo, Meriem Amer-El-Khedoud, Emma Baker, Sophie Franks, Dave Harford, Patrycja Salimi-Tabar, Laura Temple.

**Formal analysis:** Lucy Rodgers, Nicola Botting, Helen Stringer, Natalie Abdo, Meriem Amer-El-Khedoud, Emma Baker, Sophie Franks, Dave Harford, Patrycja Salimi-Tabar, Laura Temple, Ros Herman.

**Funding acquisition:** Lucy Rodgers, Nicola Botting, Ros Herman.

**Investigation:** Lucy Rodgers, Nicola Botting, Helen Stringer, Natalie Abdo, Meriem Amer-El-Khedoud, Emma Baker, Sophie Franks, Dave Harford, Patrycja Salimi-Tabar, Laura Temple, Ros Herman.

**Methodology:** Lucy Rodgers, Nicola Botting, Helen Stringer, Ros Herman.

**Project administration:** Lucy Rodgers, Natalie Abdo.

**Resources:** Lucy Rodgers.

**Software:** Lucy Rodgers.

**Supervision:** Nicola Botting, Helen Stringer, Ros Herman.

**Validation:** Lucy Rodgers, Nicola Botting, Helen Stringer, Natalie Abdo, Meriem Amer-El-Khedoud, Emma Baker, Sophie Franks, Dave Harford, Patrycja Salimi-Tabar, Laura Temple, Ros Herman.

**Visualization:** Lucy Rodgers, Ros Herman.

**Writing – original draft:** Lucy Rodgers, Nicola Botting, Helen Stringer, Natalie Abdo, Meriem Amer-El-Khedoud, Emma Baker, Sophie Franks, Dave Harford, Patrycja Salimi-Tabar, Laura Temple, Ros Herman.

**Writing – review & editing:** Lucy Rodgers, Nicola Botting, Helen Stringer, Natalie Abdo, Meriem Amer-El-Khedoud, Emma Baker, Sophie Franks, Dave Harford, Patrycja Salimi-Tabar, Laura Temple, Ros Herman.

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
