## [Decision Letter · Decision Letter 0]

Dear Dr. Rodgers,

Thank you for submitting your manuscript to PLOS ONE. After careful consideration, we feel that it has merit but does not fully meet PLOS ONE’s publication criteria as it currently stands. Therefore, we invite you to submit a revised version of the manuscript that addresses the points raised during the review process.

**Thank you for your valuable submission.**

**This is an important and well-conducted study, though some concerns need to be addressed. I encourage the authors to check all comments carefully and ensure the rebuttal clearly highlights the changes made, providing justifications for both changes and aspects left unchanged. Please respond all comments and highlight in the ms.**

**Wishing you success with the study.**

We look forward to receiving your revised manuscript.

Kind regards,

Thiago P. Fernandes, PhD

Academic Editor

PLOS ONE

**Journal Requirements:**

1. When submitting your revision, we need you to address these additional requirements. Please ensure that your manuscript meets PLOS ONE's style requirements, including those for file naming. The PLOS ONE style templates can be found at https://journals.plos.org/plosone/s/file?id=wjVg/PLOSOne_formatting_sample_main_body.pdf and https://journals.plos.org/plosone/s/file?id=ba62/PLOSOne_formatting_sample_title_authors_affiliations.pdf 2. Thank you for stating in your Funding Statement: This work was supported by the Wellcome Trust via a Health Advances in Underrepresented Populations and Diseases (HARP) Fellowship, awarded to Lucy Rodgers. Grant no. 223500/Z/21/Z.   Please provide an amended statement that declares *all* the funding or sources of support (whether external or internal to your organization) received during this study, as detailed online in our guide for authors at http://journals.plos.org/plosone/s/submit-now.  Please also include the statement “There was no additional external funding received for this study.” in your updated Funding Statement. Please include your amended Funding Statement within your cover letter. We will change the online submission form on your behalf. 3. We note that this data set consists of interview transcripts. Can you please confirm that all participants gave consent for interview transcript to be published? If they DID provide consent for these transcripts to be published, please also confirm that the transcripts do not contain any potentially identifying information (or let us know if the participants consented to having their personal details published and made publicly available). We consider the following details to be identifying information:- Names, nicknames, and initials- Age more specific than round numbers- GPS coordinates, physical addresses, IP addresses, email addresses- Information in small sample sizes (e.g. 40 students from X class in X year at X university)- Specific dates (e.g. visit dates, interview dates)- ID numbers Or, if the participants DID NOT provide consent for these transcripts to be published:- Provide a de-identified version of the data or excerpts of interview responses- Provide information regarding how these transcripts can be accessed by researchers who meet the criteria for access to confidential data, including:a) the grounds for restrictionb) the name of the ethics committee, Institutional Review Board, or third-party organization that is imposing sharing restrictions on the datac) a non-author, institutional point of contact that is able to field data access queries, in the interest of maintaining long-term data accessibility.d) Any relevant data set names, URLs, DOIs, etc. that an independent researcher would need in order to request your minimal data set. For further information on sharing data that contains sensitive participant information, please see: https://journals.plos.org/plosone/s/data-availability#loc-human-research-participant-data-and-other-sensitive-data If there are ethical, legal, or third-party restrictions upon your dataset, you must provide all of the following details (https://journals.plos.org/plosone/s/data-availability#loc-acceptable-data-access-restrictions):a) A complete description of the datasetb) The nature of the restrictions upon the data (ethical, legal, or owned by a third party) and the reasoning behind themc) The full name of the body imposing the restrictions upon your dataset (ethics committee, institution, data access committee, etc)d) If the data are owned by a third party, confirmation of whether the authors received any special privileges in accessing the data that other researchers would not havee) Direct, non-author contact information (preferably email) for the body imposing the restrictions upon the data, to which data access requests can be sent 4. Please include captions for your Supporting Information files at the end of your manuscript, and update any in-text citations to match accordingly. Please see our Supporting Information guidelines for more information: http://journals.plos.org/plosone/s/supporting-information.

Reviewers' comments:

Reviewer's Responses to Questions

**Comments to the Author**

1. Is the manuscript technically sound, and do the data support the conclusions?

Reviewer #1: No

Reviewer #2: Yes

2. Has the statistical analysis been performed appropriately and rigorously?

Reviewer #1: I Don't Know

Reviewer #2: N/A

3. Have the authors made all data underlying the findings in their manuscript fully available?

Reviewer #1: Yes

Reviewer #2: Yes

4. Is the manuscript presented in an intelligible fashion and written in standard English?

Reviewer #1: No

Reviewer #2: Yes

**Reviewer #1:**  The manuscript is significantly longer than a typical journal article, which may impact readability and adherence to journal guidelines. The content needs significant condensation, with a focus on the core findings and removal of unnecessary details. A more concise presentation of the key findings and arguments would enhance clarity and accessibility for readers. I recommend restructuring or condensing sections where possible while maintaining the integrity of the study. Consider focusing on the most essential details and streamlining discussions to improve the manuscript’s overall impact.

**Reviewer #2:**  Thank you for the opportunity to review this manuscript, which presents a valuable study on determining key intervention elements for pre-school children with co-occurring SSD and DLD. The retention of participants is impressive, and the inclusion of a diverse range of parties throughout the entire process strengthens the findings. Additionally, the clear goal of translating these results into clinical guidelines is an important contribution to the field.

However, while the paper provides useful insights, it would benefit from more specific examples and definitions of key terms and phrases. Although some readers may be familiar with the theories and concepts discussed, since this manuscript is being submitted to PLOS One rather than a speech-specific journal, I recommend that certain terms be more explicitly defined for a broader audience. Please see attached comments for more specific feedback.

I look forward to following this work.

**Do you want your identity to be public for this peer review?** For information about this choice, including consent withdrawal, please see our Privacy Policy

Reviewer #1: No

Reviewer #2: No

---

## [Author Response · Author response to Decision Letter 1]

31 Mar 2025

Editor

This is an important and well-conducted study, though some concerns need to be addressed. I encourage the authors to check all comments carefully and ensure the rebuttal clearly highlights the changes made, providing justifications for both changes and aspects left unchanged. Please respond all comments and highlight in the ms.

Response: Thank you for your feedback and for stating that this is an important and well conducted study. We have responded to both reviewer’s comments and have made changes accordingly. More detail is given under the ‘reviewer 1’ and ‘reviewer 2’ sections below.

Journal styling requirements

Response: File names checked

2. Thank you for stating in your Funding Statement. Please provide an amended statement that declares *all* the funding or sources of support (whether external or internal to your organization) received during this study, as detailed online in our guide for authors at http://journals.plos.org/plosone/s/submit-now. Please also include the statement “There was no additional external funding received for this study.” in your updated Funding Statement. Please include your amended Funding Statement within your cover letter. We will change the online submission form on your behalf.

Response: Full funding statement given in the cover letter

3. We note that this data set consists of interview transcripts. Can you please confirm that all participants gave consent for interview transcript to be published?

If they DID provide consent for these transcripts to be published, please also confirm that the transcripts do not contain any potentially identifying information (or let us know if the participants consented to having their personal details published and made publicly available). We consider the following details to be identifying information:

- Names, nicknames, and initials

- Age more specific than round numbers

- GPS coordinates, physical addresses, IP addresses, email addresses

- Information in small sample sizes (e.g. 40 students from X class in X year at X university)

- Specific dates (e.g. visit dates, interview dates)

- ID numbers

Or, if the participants DID NOT provide consent for these transcripts to be published:

- Provide a de-identified version of the data or excerpts of interview responses

- Provide information regarding how these transcripts can be accessed by researchers who meet the criteria for access to confidential data, including:

a) the grounds for restriction

b) the name of the ethics committee, Institutional Review Board, or third-party organization that is imposing sharing restrictions on the data

c) a non-author, institutional point of contact that is able to field data access queries, in the interest of maintaining long-term data accessibility.

d) Any relevant data set names, URLs, DOIs, etc. that an independent researcher would need in order to request your minimal data set.

Response: Information added to the cover letter.

Response: Captions for supporting files added to the end of the manuscript and in text citations updated to match.

Reviewer 1

The manuscript is significantly longer than a typical journal article, which may impact readability and adherence to journal guidelines. The content needs significant condensation, with a focus on the core findings and removal of unnecessary details. A more concise presentation of the key findings and arguments would enhance clarity and accessibility for readers. I recommend restructuring or condensing sections where possible while maintaining the integrity of the study. Consider focusing on the most essential details and streamlining discussions to improve the manuscript’s overall impact.

Response: Thank you for volunteering your time to kindly review our article. We have removed some content to make the paper more concise. We used the two additional supplementary materials (1.ACCORD-ACcurate COnsensus Reporting Document; 2.GRIPP- Guidance for Reporting Involvement of Patients and the Public) to inform what key information to keep in. We have also made the paper more accessible by making changes in response to reviewer 2’s comments (below).

Reviewer 2

Thank you for the opportunity to review this manuscript, which presents a valuable study on determining key intervention elements for pre-school children with co-occurring SSD and DLD. The retention of participants is impressive, and the inclusion of a diverse range of parties throughout the entire process strengthens the findings. Additionally, the clear goal of translating these results into clinical guidelines is an important contribution to the field. However, while the paper provides useful insights, it would benefit from more specific examples and definitions of key terms and phrases. Although some readers may be familiar with the theories and concepts discussed, since this manuscript is being submitted to PLOS One rather than a speech-specific journal, I recommend that certain terms be more explicitly defined for a broader audience. Please see attached comments for more specific feedback.

Response: Thank you for volunteering your time to review our paper and provide feedback. We are pleased that you found the inclusion of a diverse range of parties, retention of participants, and translation of the findings into clinical practice to be strengths. We have addressed your helpful comments relating to clarity below.

1. Consider abbreviating Speech and Language Therapist as ‘SLT’ for consistency and brevity.

Response: SLT used throughout the paper after initial acronym has been given in full in the introduction.

2. ‘Delphi’ should always be capitalized.

Response: corrected.

3. The rationale for selecting the e-Delphi methodology (lines 225–230) is well-explained, but the term ‘e-Delphi’ should be defined earlier in the manuscript, especially since it appears in the title.

Response: Thank you for feeding back that our rationale for e-Delphi methodology is well-explained.

We have added some information to the end of our introductory paragraph:

“Following the co-design of potential intervention elements with a diverse project steering group, consensus on these elements was sought using e-Delphi methodology. Within an e-Delphi, experts rate statements within an electronic survey, and statements meeting a minimum threshold for inclusion are taken forward”

4. It may be more than this paper would warrant, but it could be valuable to include a figure (such as a pie chart) to illustrate the number of key elements that reached consensus in round 1 and round 2.

Response: Thank you for this suggestion. We agree this could be a useful way of conveying the consensus rates. We are also aware of the length of the article as it is, and reviewer 1s response concerning this. You have prompted us to consider including pie charts in future presentations we give about this work-thanks again.

5. 36 & 156: Consider using either ‘a priori’ or ‘priori’

Response: ‘A priori’ added in.

6. 71–89: DLD and SSD are clearly defined and differentiated here. While the long-term outcomes are listed, it would be helpful to include specific examples of different errors associated with DLD alone, SSD alone, and the co-occurrence of DLD and SSD.

Response: Thank you for feeding back that DLD and SSD are clearly defined and differentiated.

We have provided some additional information/references regarding the errors associated with DLD/SSD co-occurring and alone:

“In contrast to children with DLD in isolation, children with phonological SSDs have diffiuclties with their expressive phonology, i.e. speech production”

“Evidence indicates that children with a co-occurring profile have a compromised linguistic system, as evidenced by their increased speech omission patterns, in contrast to features seen within a singular SSD profile”

7. 118:‘this is known as’

Response: Corrected.

8. 123-124: Could you provide examples of other approaches referenced here?

Response: Added in: “(e.g. efficacy, implementation, partnership or target-population based approaches)”

9. 127: The term ‘stakeholders’- is this referring to professionals and individuals with lived experience? Are others included in this group? Could you clarify if this is analogous to the ‘steering group’ mentioned earlier?

Response: Yes this is analogous to the steering group. For clarity, we have taken out the term stakeholder here (which may refer to a wider range of people), and specified:

“whereby relevant professionals and people with lived experience are involved in the co-design of the intervention”

10. 154: The phrase ‘to date’ – is this referring to ongoing research?

Response: The e-Delphi is phase 3 in a 4 phase intervention development body of work. We have added detail in to help clarify:

• “This study builds on this pre-study PPI and CE, as a part of a four-phase body of work…”

• “Phases one and two included…”

• “The current study, phase 3,…”

11. 179: ‘Recuitment’ – please check spelling

Response: Corrected.

12. 209: What were the specific probes used? Would it be possible to provide examples?

Response: We have an example of a probe within supplementary materials 3 (S3 Pre-design steering group activities) but we needed to make this clearer. We have therefore added to the text:

“both of which are included in S3. Pre-design steering group activities”

13. 217: It may be helpful to spell out ‘BCT’ in the text, even if it’s already defined in table 1

Response: BCT spelt out again in the text.

Individual meetings involved the identification of potential “Behaviour Change Techniques (BCTs) …. “

14.295-305: Approximately how long did the survey take to complete? What percentage left free text comments?

Response: Unfortunately, the response Metadata on Qualtrics includes any time that the panellists had the survey open but not necessarily working on it (they had the option to complete some of the survey and come back to it later/leave it open on their browser). However, the piloting process indicated that the survey took between 45 mins to 1 hour to complete, and we have added this in:

“The piloting process indicated that it would take between 45 minutes and an hour to complete the survey.”

“Although free text comments were optional, 86% (31/36) of panellists left a free text comment for one or more statement.”

15. 393-395: The number of key elements by group is useful, but adding percentages could provide more clarity.

Response: Percentages added

“Following the second round, consensus had been reached on 44 (100%) intervention elements (statements) in total; 10 core elements relating to intervention targets (22.7%), 18 core elements relating to intervention content (40.9% ), 8 core elements relating to intervention delivery (18.2% ), 4 core elements relating to the intervention manual (9.1% ), and 4 core elements regarding future adaptions (9.1% ).”

16. 407: Is ‘within the research arena’ supposed to be ‘area’?

Response: Changed to ‘within research’.

17. 434: ‘heterogenous’-please check spelling

Response: Corrected.

18. 473: ‘will need to be addressed”

Response: Corrected.

19. Page 10: Please cite Miro similarly to how you cite Qualtrics. Alternatively, add ‘whiteboard’ whenever Miro is mentioned.

Response: In text Miro changed to “Miro ©”

Whiteboard stated after referring to Miro e.g “…. opportunity to add/expand on the Miro © whiteboard.”

20. Pages 10 & 11: How were the sub-groups determined? Could you clarify who was in each group?

Response: Added in-

“*Allocation to sub-groups varied between meetings. However, to ensure a spread of expertise, sub-groups were devised so that they each contained a person with lived experience (DH or SF) and a professional with equality, diversity and inclusion (EDI) expertise (M A-K or P S-T) where possible.”

21. Page 20: It seems like ‘to’ or ‘with’ is missing here: ‘By agreeing to/with the child’s targets’

Response: We have put ‘agree the child’s targets’ as putting ‘agree to/with the child’s targets’ may be interpreted as the targets have been identified by the SLT alone, implying a passive role by the parent.

22. Page 25: ‘Withdrawing cues will enable to the parent to’

Response: Corrected.

23. Page 27: ‘this will make the experience more positive for the parent and support with motivation.’ – should this be ‘support them with motivation,’ ‘support their motivation,’ or ‘support motivation’? (similar wording on Pages 31 & 43)

Response: Corrected to ‘support motivation’.

24. Page 29: ‘By monitoring how they are getting on, the parent will be able to modify their input accordingly.’

Response: Corrected.

25. Page 37: ‘Intervention techniques and activities will be explicitly linked to the target (or target area?) they area addressing.’ (similar phrasing on Page 42)

Response: Corrected to page 47 ‘they are’.

26. Pages 38, 39, 40, & 45: Please be consistent with ‘face-to-face.’ (also Line 468)

Response: All face-to-face for consistency.

---

## [Decision Letter · Decision Letter 1]

Dear Dr. Rodgers,

Thank you for submitting your manuscript to PLOS ONE. After careful consideration, we feel that it has merit but does not fully meet PLOS ONE’s publication criteria as it currently stands. Therefore, we invite you to submit a revised version of the manuscript that addresses the points raised during the review process.

Please respond all comments and highlight them in the revised ms.

We look forward to receiving your revised manuscript.

Kind regards,

Thiago P. Fernandes, PhD

Academic Editor

PLOS ONE

Journal Requirements:

Reviewers' comments:

Reviewer's Responses to Questions

**Comments to the Author**

Reviewer #2: (No Response)

Reviewer #3: (No Response)

Reviewer #4: (No Response)

2. Is the manuscript technically sound, and do the data support the conclusions?

Reviewer #2: Yes

Reviewer #3: Yes

Reviewer #4: Yes

3. Has the statistical analysis been performed appropriately and rigorously?

Reviewer #2: Yes

Reviewer #3: Yes

Reviewer #4: N/A

4. Have the authors made all data underlying the findings in their manuscript fully available?

Reviewer #2: Yes

Reviewer #3: Yes

Reviewer #4: Yes

5. Is the manuscript presented in an intelligible fashion and written in standard English?

Reviewer #2: Yes

Reviewer #3: Yes

Reviewer #4: Yes

Reviewer #2: Thank you for the thorough revisions. The manuscript now provides greater clarity in its framing and use of terminology, and the expanded methodological detail strengthens the transparency of the study. I particularly appreciate the added context regarding the role of stakeholders and the clarification of the e-Delphi process.

At this stage, I have only a few remaining suggestions. These are included in the Reviewer Attachments. This study makes an important contribution to the literature on intervention development for children with co-occurring SSD and DLD, and I look forward to seeing it published and informing future clinical work.

Reviewer #3: Thank you for the opportunity to review this work! The development of intervention programs targeting both speech and language is a valuable and much-needed contribution to the field. Overall, this is a compelling project, and I appreciated the opportunity to read it. I’ve noted several areas where further clarification or editing may be helpful, and I hope these comments support your efforts to strengthen the manuscript even further.

Line 84: You introduce the term ‘phonological SSDs’. It would be helpful to clarify whether you are distinguishing between phonological and articulation disorders throughout the paper, or simply noting that SSDs can have a phonological component. As you know, many children with SSDs present with both articulatory and phonological impairments. This distinction is important, as it was unclear at times whether references to SSDs throughout the manuscript were meant to include all types or specifically phonological SSDs. For instance, at Line 237, it’s not clear whether panelists were instructed to consider only phonological SSDs when reviewing the statements. In the discussion (Line 433), you refer to children with “consistent phonological SSD”, a term not previously defined or used in the paper. The manuscript would benefit from a clear statement early on about whether your focus is exclusively on children with consistent phonological disorder. If so, consider assigning a specific term (e.g., P-SSD) to differentiate it from SSDs more broadly, and revise relevant sections in the introduction, methods, and discussion accordingly.

Line 85: As currently written, the sentence suggests that speech comprehensibility is defined as “frustrations at not being understood by others around them,” which is somewhat misleading. A clearer definition of speech comprehensibility would benefit the reader. It may also be useful to briefly contrast comprehensibility with intelligibility, since intelligibility typically refers to the accuracy of speech sounds, whereas comprehensibility relates more to the listener’s ability to derive meaning from the overall message. Additionally, clarifying why comprehensibility was chosen over intelligibility in this context would strengthen this section.

Line 95: The phrase “referred to in day-to-day practice” is a bit vague, and the sentence reads as somewhat long. If your point is that these protocols are not commonly implemented, a clearer version might be: “However, clinical research suggests that intervention protocols are not strictly implemented in day-to-day practice. Instead, speech and language therapists…”

Line 98: It would be helpful to include an example to illustrate what is meant by the “infeasibility of delivery in public services” and the competing priorities clinicians face. This is a strong point in your argument, emphasizing the importance of including SLTs in intervention development, and could be further strengthened with a more detailed description of these barriers.

Line 156: Did you mean “a priori” here? If so, consider adjusting the phrasing for accuracy.

Lines 183–190: I commend you for assembling such a diverse team of contributors. Including both a parent of a child with DLD and an adult with DLD adds valuable perspectives that enrich the development process.

Table 1: Is Miro board widely known among your intended readership? This may just be a lack of knowledge on my part, but I had to look it up myself, so it may be helpful to briefly explain what it is or why it was used, or use a more common term (whiteboard) that describes the same idea.

Line 340: You mention 36 panelists but describe the main employer for only 35. Could you clarify what information is

available for the remaining participant?

Table 2, Row 1: Should this be phrased as “agree on the child’s targets” and “by agreeing on the child’s targets”? It may be a dialect difference, but I found the current wording slightly unclear.

Table 2, Statement 26: There appears to be a typo in the rationale: “enable to parent” should likely be “enable the parent.”

Table 2, Statement 20: This is the first use of the abbreviation SLT, which I don’t believe has been defined earlier in the paper. Consider introducing the term when it first appears and using throughout, or changing this instance.

Table 2, Statement 47: This is the only statement formatted with “Statement 47” preceding the text, whereas others list only the number in parentheses.

Table 3, Row 1: I believe there is a typo in the reworded statement where “area” should likely be “are.”

Table 1 (Behavior Change Techniques): The first mention of behavior change techniques and behavior change technique ontology appears in the Methods section. Since these are central to your intervention design, consider introducing them earlier in the Introduction to help the reader understand their significance.

Line 421: When discussing the tension between service delivery and feasibility, could you specify which panelist responses reflected this tension? Providing explicit examples would strengthen your argument.

Line 423: The phrase “flexibility relating to deliverer” is unclear. Do you mean flexibility regarding how intervention is delivered or the type of intervention, or? A more explicit phrasing would improve comprehension.

Line 447: Could you clarify which specific findings you are referring to here? Although I understand what you are getting at, being more explicit would make this section more impactful and easier to follow.

Line 462: You note both a change in how statements were worded and a shift in how participants were asked to consider them (e.g., setting aside limitations of their own service). How can you be confident that the change in responses was due to the instruction rather than the rewording itself? It may be worth addressing this potential confound.

Line 494: Although your phrasing of this limitation is acceptable, I would argue that the limitation lies less in the number of panelists (as you reached your recruitment target) and more in the uneven representation of different service delivery models among them.

Reviewer #4: Thank you for the opportunity to review this important work. The stated purpose of this study was to achieve consensus on core intervention approaches that are specific to DLD and SSD. The research question is well motivated and justified and the findings make a significant contribution to the literature surrounding intervention approaches in children who have concomitant speech and language difficulties. This work is especially important in light of the CATALISE reclassification of DLD, which highlights phonology as an important interface between DLD and SSD. As a DLD researcher myself, the relation between DLD and SSD is understudied, so this work advances an important topic of discussion. In my view, this work is well positioned at the crux of a high-impact question that is of great interest to the field. I commend the authors on tackling this important question and believe this work will be of great interest to the readership of this journal. While this is more specific to speech and language intervention, the types of intervention components are translatable to multiple clinical populations and the Delphi method is well explained. The methodological detail, while lengthy, provides a useful model for researchers following the Delphi method to conduct systematic and rigorous inquiry into clinical questions. In light of my examination of the previous reviews and author responses, my comments are minor and are offered to improve readability and add to this important question.

Introduction:

Between lines 84-121 the authors discuss shared deficits in phonology between DLD and SSD. In my view, this section would be enhanced with a few statements on some of the research related to the mechanistic ties that might support overlapping areas of deficits. What comes to mind is recent work in sequence learning in DLD by Goffman & Gerken (2023), Benham et al., (2018), and Benham & Goffman (2022) To an extent, the procedural deficit hypothesis (Ullman & Pierpont, Ullman & Pullman) may also be a useful rationale for bridging these areas. While the work in this manuscript is not theoretical in nature, a brief discussion of some of this work may provide a useful framework for motivating the ‘theory-based intervention approaches’ you address below. Some recommended citations for consideration are:

Goffman, L., & Gerken, L. (2023). A developmental account of the role of sequential dependencies in typical and atypical language learners. Cognitive Neuropsychology, 40(5-6), 243-264.

Benham, S., Goffman, L., & Schweickert, R. (2018). An application of network science to phonological sequence learning in children with developmental language disorder. Journal of Speech, Language, and Hearing Research, 61(9), 2275-2291 .

Benham, S., & Goffman, L. (2022). A longitudinal study of the phonological organisation of novel word forms in children with developmental language disorder. International journal of speech-language pathology, 24(2), 212-223.

Methods:

The authors use the term “Behaviour Change Technique” quite a bit, but I found this lacking a definition. Some examples would also be helpful. It would be helpful to define this around line 229 so readers can understand why this is a core part of the intervention process.

Table 2 would benefit from having the number of the statement on the side to the left so that it is easier to refer back to.

On Table 3, it would be useful to add the values of consensus to the table instead of just reporting that no consensus was reached. It would be of interest to know how far off the agreement was for these statements.

The rate of attrition is impressive. Did the participant who dropped out provide a reason or just did not respond to the round 2 survey?

Discussion

I am curious about the distinction the authors make between articulatory-based deficits and phonological-based speech sound deficits. I find that this line can often be exceptionally vague, but this work seems to take an approach that is broad and can be applied to a number of SSDs. I think this section would benefit from a brief discussion of the application of these statements to various SSDs around line 436 when the authors mention consistent phonological SSDs. Often children with CAS, who are less consistent, also exhibit language deficits, so I am curious whether these statements would be applicable to special populations who exhibit less consistency.

**Do you want your identity to be public for this peer review?** For information about this choice, including consent withdrawal, please see our Privacy Policy

Reviewer #2: No

Reviewer #3: No

Reviewer #4: No

---

## [Author Response · Author response to Decision Letter 2]

19 May 2025

REVIEWER 2

Author’s response: We are pleased that you found the added context and clarification of the e-Delphi process helpful. Thank you once again for your helpful feedback, we have addressed your comments below.

1.Thank you for clarifying the use of Miro whiteboards. However, is it necessary to name Miro specifically, or would a more general term such as “online whiteboard” suffice for the purposes of the Methods section?

-We have replaced with “online whiteboard” (table 1)

2.Were panellists offered any form of compensation for their participation?

-They were not. However, they were given the option to have their name put in the acknowledgements section of the paper if they wished.

3. Line 293: Could you clarify the rationale behind selecting 50% as the threshold for reconsideration?

-Great question. There is no “set” recommended cut off point within Delphi study guidance (Diamond et al, 2014; Trevelyan et al, 2015). We opted for 50% or higher (rather than complete exclusion) as these statements had shown some acceptability amongst the panellists and therefore had greater potential for increased appropriateness subject to re-wording.

4. Lines 175–179; 201–206: Consider condensing each set of bullet points into a single sentence.

-We have kept 175-179 (now lines 204-208) as it is as we feel that the separate numbered points clearly signpost the reader to the upcoming two separate sections (co-design + delphi) in the methodology. We have combined the bullet points into a single sentence in 201-206 (now lines 230-232)

5. Table 2: Consider simplifying table 2 to include just two columns- statement and rationale. The accompanying percentage, median, and IQR data could be moved to the supplementary materials. While informative, this quantitative detail may be redundant in the main text since the paper already defines agreement as <75%, and the range of agreement percentages is mentioned prior to the table.

- Thank you for this suggestion. Although we understand what you mean, we feel it best to leave this information on the table so that it is readily available for the reader to access and understand to what level of agreement each statement reached. This is like other tables from e-Delphi studies which have been published in the journal e.g. Curtin et al. (2024) https://journals.plos.org/plosone/article?id=10.1371/journal.pone.0301722#sec038

6. Table 3: This is a strong and well-constructed table. However, as the primary focus of the paper is the development of the new scale, consider moving Table 3 to the Supplementary Materials. It appears to be more relevant to the Methods and may not be essential in the main body of the manuscript.

- We understand your point and have moved this to the supplementary materials (S7 Reworded statements)

REVIEWER 3

Author’s response: We are thrilled that you find this to be a compelling project and a much-needed contribution to the field. Thank you so much for taking the time to review our paper. We have addressed your helpful comments in the table below.

1.Line 84: You introduce the term ‘phonological SSDs’. It would be helpful to clarify whether you are distinguishing between phonological and articulation disorders throughout the paper, or simply noting that SSDs can have a phonological component. As you know, many children with SSDs present with both articulatory and phonological impairments. This distinction is important, as it was unclear at times whether references to SSDs throughout the manuscript were meant to include all types or specifically phonological SSDs. For instance, at Line 237, it’s not clear whether panelists were instructed to consider only phonological SSDs when reviewing the statements. In the discussion (Line 433), you refer to children with “consistent phonological SSD”, a term not previously defined or used in the paper. The manuscript would benefit from a clear statement early on about whether your focus is exclusively on children with consistent phonological disorder. If so, consider assigning a specific term (e.g., P-SSD) to differentiate it from SSDs more broadly, and revise relevant sections in the introduction, methods, and discussion accordingly.

- Added in: “In this paper (and study) we differentiate between phonology and articulation, as articulation refers to distorted sound productions (e.g. a lisp) which respond best to traditional articulation therapy rather than cognitive-linguistic approaches” (lines 88-91)

- Added in: “Intervention approaches for children with P-SSD in isolation varies according to their error characteristics. For example, unlike children with consistent P-SSD errors, children with inconsistent P-SSD errors respond best to a core vocabulary approach. This variation in response to intervention has implications for intervention development, as targeting children with both consistent and inconsistent errors will likely impact intervention specificity. This study focuses on children with consistent SSD errors, who make up the largest proportion of children presenting within P-SSD” (lines 107-113)

- The term P-SSD has been added throughout the paper.

2. Line 85: As currently written, the sentence suggests that speech comprehensibility is defined as “frustrations at not being understood by others around them,” which is somewhat misleading. A clearer definition of speech comprehensibility would benefit the reader. It may also be useful to briefly contrast comprehensibility with intelligibility, since intelligibility typically refers to the accuracy of speech sounds, whereas comprehensibility relates more to the listener’s ability to derive meaning from the overall message. Additionally, clarifying why comprehensibility was chosen over intelligibility in this context would strengthen this section

- Added in: “Speech comprehensibility refers to listener being able to derive meaning from what a child has communicated based on their speech and wider contextual cues. This contrasts to the term speech intelligibility where meaning is more closely linked to the accuracy of the child’s speech sound production15. The current study is concerned with comprehensibility rather than intelligibility, due to the knock on impacts on the child’s everyday life. For example, the adults around them not being able to understand their needs, leading to frustration” (lines 97-103)

3.Line 95: The phrase “referred to in day-to-day practice” is a bit vague, and the sentence reads as somewhat long. If your point is that these protocols are not commonly implemented, a clearer version might be: “However, clinical research suggests that intervention protocols are not strictly implemented in day-to-day practice. Instead, speech and language therapists…"

- Rephrased: “However, clinical research highlights that intervention protocols are often not strictly implemented within clinical practice. Instead, speech and language therapists (SLTs) ‘mix and match’…” (lines 116-118)

4.Line 98: It would be helpful to include an example to illustrate what is meant by the “infeasibility of delivery in public services” and the competing priorities clinicians face. This is a strong point in your argument, emphasizing the importance of including SLTs in intervention development, and could be further strengthened with a more detailed description of these barriers.

- Rephrased and added to: “Reasons for this might include infeasibility of delivery in public services, which is one consequence of implementation factors, such as the views and experiences of SLTs, not being studied within intervention development processes. Infeasibility within public services may relate to the aims (outcomes) of intervention research not necessarily reflecting what SLTs would prioritise within their practice, or the complex, clinical profiles of the children they support. The active involvement of SLTs within the development of new interventions therefore has the potential to mitigate these issues” (lines 120-127)

5. Line 156: Did you mean “a priori” here? If so, consider adjusting the phrasing for accuracy.

- Rephrased to “prior” (line 191)

6.Lines 183–190: I commend you for assembling such a diverse team of contributors. Including both a parent of a child with DLD and an adult with DLD adds valuable perspectives that enrich the development process.

- Thank you. We have been incredibly fortunate to have such a diverse and reflective team.

7.Table 1: Is Miro board widely known among your intended readership? This may just be a lack of knowledge on my part, but I had to look it up myself, so it may be helpful to briefly explain what it is or why it was used, or use a more common term (whiteboard) that describes the same idea.

- We have taken “Miro” out and referred to it as an “online whiteboard” instead (table 1)

8.Line 340: You mention 36 panelists but describe the main employer for only 35. Could you clarify what information is

available for the remaining participant?

- Well spotted- thank you! This was a typo, there were 5 panellists from independent practice, not 4 (line 373).

9.Table 2, Row 1: Should this be phrased as “agree on the child’s targets” and “by agreeing on the child’s targets”? It may be a dialect difference, but I found the current wording slightly unclear.

- Interesting, thank you. We do wonder if this is a dialectal difference. The clarity of this statement didn’t come up as an issue in the piloting. Although we understand your point we would rather keep the statement as it is, just because this is how we phrased it in the e-Delphi survey itself.

10. Table 2, Statement 26: There appears to be a typo in the rationale: “enable to parent” should likely be “enable the parent.”

-Corrected, table 2

11.Table 2, Statement 20: This is the first use of the abbreviation SLT, which I don’t believe has been defined earlier in the paper. Consider introducing the term when it first appears and using throughout, or changing this instance.

- SLT definition added in, table 2

12. Table 2, Statement 47: This is the only statement formatted with “Statement 47” preceding the text, whereas others list only the number in parentheses.

- In response to reviewer 3’s comments, we have now changed the table so that the statement number is given before each statement (not after in parentheses).

13. Table 3, Row 1: I believe there is a typo in the reworded statement where “area” should likely be “are.”

- Corrected. Please note that table 3 is now in the supplementary materials (S7 Reworded statements)

14.Table 1 (Behavior Change Techniques): The first mention of behavior change techniques and behavior change technique ontology appears in the Methods section. Since these are central to your intervention design, consider introducing them earlier in the Introduction to help the reader understand their significance.

- Added in: “Additionally, behaviour change theory can be used to define how a SLT might a parent in carrying out activities at home; for example, the SLT might give the parent specific feedback on how they are performing a linguistic technique with their child in clinic. Explicitly specifying BCTs is important for future study replicability, as well as for our understanding of which BCTs may be ‘active ingredients’ within the intervention (i.e. essential to the intervention being a success). The behaviour change technique ontology (BCTO) provides a standard terminology for BCTs, for use within the development and evaluation of complex interventions”(lines 139-146)

15. Line 421: When discussing the tension between service delivery and feasibility, could you specify which panelist responses reflected this tension? Providing explicit examples would strengthen your argument.

- Added in: “The panellist responses reflected this tension. Round one showed that some panellists were strongly against parents not doing speech related activities at home, whereas other panellists strongly agreed that it should be the role of the SLT to provide this support in clinic. Consensus was reached….” (lines 455-458)

16. Line 423: The phrase “flexibility relating to deliverer” is unclear. Do you mean flexibility regarding how intervention is delivered or the type of intervention, or? A more explicit phrasing would improve comprehension.

- Rephrased: “Flexibility relating to deliverer (e.g. by the SLT, parent, or both) is not routine across…” (lines 459-460)

17. Line 447: Could you clarify which specific findings you are referring to here? Although I understand what you are getting at, being more explicit would make this section more impactful and easier to follow.

- Added in: “Therefore, the updated (psycholinguistic) target area and option of corresponding intervention techniques (e.g. syllable segmentation, speech recasting) have been kept deliberately flexible….” (lines 474-476)

18. Line 462: You note both a change in how statements were worded and a shift in how participants were asked to consider them (e.g., setting aside limitations of their own service). How can you be confident that the change in responses was due to the instruction rather than the rewording itself? It may be worth addressing this potential confound.

- Rephrased: “There is no guarantee that the consensus of re-worded statements in round 2 was a direct result of panellists disregarding their service limitations in their responses. However, free text comments in round 1 highlighted a clear discrepancy between how SLTs might ideally carry out their practice and how this is actualised within the context of service limitations” (lines 484-488)

19. Line 494: Although your phrasing of this limitation is acceptable, I would argue that the limitation lies less in the number of panelists (as you reached your recruitment target) and more in the uneven representation of different service delivery models among them.

- Rephrased: “One potential limitation of the e-Delphi was the representation of panellists and resulting limitations to statistical analysis. A recruitment target…” (line 541 )

REVIEWER FOUR

Author’s response: We are so pleased to hear that you find our work to be of high impact and appreciate the time you have taken to review our paper and provide your very helpful comments. We have addressed them in the table below.

1) Between lines 84-121 the authors discuss shared deficits in phonology between DLD and SSD. In my view, this section would be enhanced with a few statements on some of the research related to the mechanistic ties that might support overlapping areas of deficits. What comes to mind is recent work in sequence learning in DLD by Goffman & Gerken (2023), Benham et al., (2018), and Benham & Goffman (2022) To an extent, the procedural deficit hypothesis (Ullman & Pierpont, Ullman & Pullman) may also be a useful rationale for bridging these areas. While the work in this manuscript is not theoretical in nature, a brief discussion of some of this work may provide a useful framework for motivating the ‘theory-based intervention approaches’ you address below. Some recommended citations for consideration are: Goffman, L., & Gerken, L. (2023). A developmental account of the role of sequential dependencies in typical and atypical language learners. Cognitive Neuropsychology, 40(5-6), 243-264. Benham, S., Goffman, L., & Schweickert, R. (2018). An application of network science to phonological sequence learning in children with developmental language disorder. Journal of Speech, Language, and Hearing Research, 61(9), 2275-2291. Benham, S., & Goffman, L. (2022). A longitudinal study of the phonological organisation of novel word forms in children with developmental language disorder. International journal of speech-language pathology, 24(2), 212-223.

- We found these papers very interesting, thank you for sharing them. We have added some information gleaned from the papers into our introduction and have bookmarked the papers for an article we will be developing soon about the intervention theory.

Added in: “Recent research using network science has highlighted phonological sequence production as a unique area of difficulty in children with DLD, providing further evidence for the link between P-SSD and DLD” (lines 95-97)

Added in: “Recent research suggests that this

---

## [Editor Report · Decision Letter 2]

Co-design to consensus: identifying the core elements of a novel intervention for pre-school children with co-occurring phonological speech sound disorder (SSD) and developmental language disorder (DLD) using a modified e-Delphi approach

PONE-D-24-58856R2

Dear Dr. Rodgers,

We’re pleased to inform you that your manuscript has been judged scientifically suitable for publication and will be formally accepted for publication once it meets all outstanding technical requirements.

Kind regards,

Thiago P. Fernandes, PhD

Academic Editor

PLOS ONE

Additional Editor Comments (optional):

Thank you for your careful and thoughtful edits. The ms reads much better now - but please double check grammar and refs list to help streamline typesetting.

Wishing you success with the study.
---

## [Editor Report · Acceptance letter]

PONE-D-24-58856R2

PLOS ONE

Dear Dr. Rodgers,

I'm pleased to inform you that your manuscript has been deemed suitable for publication in PLOS ONE. Congratulations! Your manuscript is now being handed over to our production team.

Kind regards,

on behalf of

Dr. Thiago P. Fernandes

Academic Editor

PLOS ONE